# What Can We Learn from the Previous Research on the Symptoms of Selective Mutism? A Systematic Review

**DOI:** 10.3390/bs15111485

**Published:** 2025-10-31

**Authors:** Judith Kleinheinrich, Felix Vogel

**Affiliations:** 1Department of Child and Adolescent Psychotherapy, University of Hamburg, 20146 Hamburg, Germany; 2Department of Clinical Psychology and Psychotherapy of Childhood and Adolescence, Johannes Gutenberg University, 55131 Mainz, Germany

**Keywords:** selective mutism, children, anxiety, symptoms, systematic review, assessment, diagnostics

## Abstract

Accurate understanding of a mental disorder’s symptomatology is essential for valid diagnosis, differential assessment, and treatment planning. It is therefore remarkable that failure to speak is defined as the only symptom in the diagnostic criteria of selective mutism (SM) in current classification systems. This narrow definition may not reflect the full range of difficulties experienced by affected children. This systematic review aimed to synthesize empirical findings on the broader symptomatology of SM across diverse study designs, informants, and assessment methods. Following PRISMA guidelines, we searched PubMed, Web of Science, and APA PsycNet, leading to 82 studies with participant samples (beyond single case reports) included in the final analysis. Results indicated that social and unspecific anxiety were the most frequently assessed and consistently identified symptoms. However, additional features—including withdrawal, depressive symptoms, social skill deficits, and, in qualitative accounts, externalizing and oppositional behaviors—were also documented. The observed symptom diversity varied notably across assessment methods and informants. Our findings support a multisymptomatic understanding of SM and suggest that failure to speak alone do not fully account for its clinical presentation. A more differentiated conceptualization may enhance diagnostic precision, inform individualized intervention strategies, and contribute to discussions on refining diagnostic frameworks.

## 1. Introduction

Selective mutism (SM) is a disorder characterized by a consistent failure to speak in specific social situations in which there is an expectation for speaking (e.g., at school), despite the ability to speak in other contexts (e.g., at home with family members; [2]). According to the latest edition of the Diagnostic and Statistical Manual of Mental Disorders (DSM-5), this failure to speak must persist for at least one month (excluding the first month of school), interfere with educational or social functioning, and cannot be attributed to a lack of familiarity with the spoken language or to another communication or developmental disorder ([2]). The disorder typically manifests during early childhood ([2]; [20]; [54]), with point prevalence estimates ranging from 0.03% to 1% ([2]; [34]). The theoretical conceptualization of SM has evolved significantly over time. Historically, SM was referred to as elective mutism, a term that implied a volitional refusal to speak, often interpreted as a sign of defiance or oppositional behavior ([48]). This perspective has shifted substantially over the last decades in light of growing evidence of strong associations between SM and anxiety ([6]; [19]; [34]; [48]; [57]) In line with these findings, SM has been reclassified as an anxiety disorder in the DSM-5 ([2]). However, this conceptualization had not gone unchallenged. A recent meta-analysis by [19] ([19]) found that while 80% of children with SM were diagnosed with an additional anxiety disorder, notably social phobia, considerable heterogeneity was present across studies. Importantly, their findings suggest that some children with SM may not exhibit clinically significant anxiety symptoms, indicating that other underlying mechanisms may be relevant for a subgroup of affected children. These findings align with discussions in the literature regarding alternative explanatory approaches to SM ([19]; [35]; [34]), suggesting that the etiology and symptom profile of SM may be more heterogeneous than currently reflected in diagnostic frameworks.

SM can have serious and lasting consequences for affected children. Research shows the disorder’s high persistence and generally unfavorable prognosis when not properly identified or adequately addressed ([5]; [25]; [41]). Left unrecognized or misunderstood, SM can affect children’s functionality in social, academic, and psychological domains ([9]; [14]; [45]), often resulting in affected children performing below average levels at school ([31]) and experiencing substantial deficits in the development of social skills. Children with SM consistently demonstrate lower levels of social assertion, self-control, and overall social competence compared to their peers, along with reduced self-esteem ([9]; [13]; [41]). These deficits lead to ongoing difficulties in peer interactions and academic settings ([5]; [32]). Beyond such functional impairments, the uncertainty and psychological stress caused by the inability to speak in expected situations represent a significant burden in themselves. If SM remains untreated, its effects can persist into adulthood. Longitudinal studies report ongoing communication challenges and an elevated risk for secondary mental health conditions, particularly mood and anxiety disorders ([41]). Adults with a history of SM exhibit lower self-esteem, increased interpersonal anxiety, and significantly impaired communication skills compared to healthy controls ([53]). Furthermore, they frequently display diminished independence, motivation, self-confidence, and emotional maturity, as well as a decreased ability to cope with stress ([41]).

Research indicates that the chances of remission increase when treatment is initiated early in the disorder’s course ([25]; [36]; [62]). Early intervention improves outcomes but also reduces the risk of secondary complications and the emergence of avoidance behaviors that can become increasingly challenging to address over time. Early recognition therefore facilitates the application of suitable therapeutic approaches before maladaptive behaviors become ingrained and secondary complications develop.

For accurate and early identification of SM, comprehensive knowledge about the condition’s symptomatology is essential. Understanding the symptoms of mental disorders not only informs clinical decision-making and intervention planning but also provides the basis for legitimate and valid diagnostic classifications.

However, the current diagnostic criteria in both the DSM-5 and the International Classifications of Diseases (ICD-10; [61]) identify failure to speak itself as the core symptom of SM, with additional behavioral, emotional, or physiological features not formally defined. This narrow diagnostic frame stands in contrast to a growing body of research suggesting that SM may be a multisymptomatic disorder ([11]; [23]; [34]; [59]). This discrepancy is particularly striking given that the DSM-5 itself defines mental disorders as syndromes characterized by clinically significant disturbances across multiple domains of cognition, emotion, or behavior ([2]). While the DSM-5 does list additional clinical features within its descriptive sections for SM–such as excessive shyness, social anxiety, and withdrawal–these aspects appear as associated characteristics rather than formal diagnostic criteria. As a result, formal diagnosis is limited to the identification of situational failure to speak alone, reducing a potentially complex multisymptomatic condition to a single behavioral indicator. This raises important questions about whether current diagnostic frameworks sufficiently capture the full clinical presentation of SM and whether they offer adequate guidance for comprehensive evaluation and targeted intervention ([59]).

While failure to speak is undoubtedly the most visible and defining feature of SM, research has long suggested that this behavior may not exist in isolation. Rather, it may be embedded within a broader range of symptoms ([10]; [23]; [44]). However, only few studies have explicitly examined the multifaceted symptomatology of SM. Among these, one study directly asked parents to report symptoms beyond failure to speak in an open-ended format ([59]). The results indicated that the majority of children displayed various symptoms in addition to mutism, including fear, motor freezing, and avoidance behavior, implying that SM frequently presents with a complex, heterogeneous clinical profile. Beyond these systematic insights, additional symptoms have been documented in clinical reports and observational studies. Children with SM have frequently been characterized as anxious and withdrawn ([21]; [51]), and behavioral signs such as freezing, reduced social skills, or psychomotor inhibition have been noted in structured assessments ([32]; [56]; [64]).

Even though certain symptom clusters are consistently documented, the clinical presentation of SM remains highly variable across studies. This heterogeneity might in part be due to methodological variation in assessment and reporting of symptoms. Several studies have demonstrated that informant perspectives can diverge considerably even within SM populations. For instance, while parents and teachers often report internalizing symptoms such as withdrawal or anxiety, children with SM tend to underreport such difficulties ([9]; [29]). Moreover, parents have been found to rate symptoms such as social withdrawal more severely than teachers ([24]). These discrepancies suggest that both the source of information and the choice of assessment method (e.g., questionnaires, open inquiries or behavioral observations) may influence which symptoms are captured and how they are interpreted.

Although previous reviews have addressed selected aspects of SM–such as etiology, comorbidities, or treatment efficacy ([19]; [22]; [34]; [50])—to our knowledge, no systematic review has comprehensively examined the full range of symptoms reported in children with SM across different study designs, assessment methods, and informants. This constitutes a critical gap in the literature, as a more nuanced understanding of the disorder’s symptom profile is essential for effective clinical practice and individualized treatment.

The present review seeks to address this limitation by systematically identifying and synthesizing the symptoms of SM reported in empirical studies involving children and adolescents. A particular focus is placed on how symptoms are assessed and reported across different data collection methods (e.g., questionnaires, interviews, observations) and informants (e.g., caregivers, teachers, affected children). By examining patterns in symptom reporting, this study aims to clarify which symptoms are consistently observed across contexts and which may be more method- or informant-dependent.

The rationale for this review is twofold. First, as outlined above, accurate and thorough knowledge of a disorder’s symptomatology is fundamental for diagnostic validity–especially for conditions like SM, in which symptoms may be situational, subtle, or selectively expressed. Second, a richer understanding of symptom patterns can inform differential diagnosis (e.g., distinguishing SM from social anxiety disorder or autism spectrum disorder) and guide the development of more individualized and effective treatment strategies. A systematic synthesis of symptoms may also provide a basis for future efforts to subtype SM ([10]; [17]; [33]).

By providing an integrative overview of the empirical evidence on SM symptoms, this review aims to support a more differentiated conceptualization of the disorder and to inform both clinical practice and future research.

## 2. Materials and Methods

This review was conducted in accordance with the Preferred Reporting Items for Systematic Reviews and Meta-Analyses (PRISMA) guidelines ([38]). The protocol for this review was registered in the Open Science Framework (OSF) Registries under the registration doi https://doi.org/10.17605/OSF.IO/RHV42.

### 2.1. Search Strategy

A comprehensive literature search was performed to identify all empirical studies investigating symptoms associated with SM. The electronic databases PubMed, Web of Science, and APA PsycNet were searched. This combination was selected to ensure coverage of both medical and psychological literature, thereby maximizing the likelihood of identifying all relevant publications. The search was conducted through 24 February 2025, using the search terms “selective mutism” and “elective mutism”. As these terms have historically defined the disorder, their inclusion was considered sufficient to capture the relevant literature base, consistent with previous meta-analyses in this field ([19]; [26]). Search parameters varied across databases: no restrictions were set in PubMed and APA PsycNet, whereas Web of Science was limited to studies using the filter “Document Type = Article”. Each search was documented, including the research date, database searches, and number of records retrieved.

### 2.2. Study Selection Procedure

All identified references were imported into Zotero (Version 7.0.16; [52]), a reference management system, for organization and duplicate management. Duplicate records were automatically detected and subsequently removed through manual verification to ensure accuracy.

The selection process followed a two-stage screening approach. Initially, all unique records were evaluated based on titles and abstracts by a single reviewer. Studies that clearly did not meet inclusion criteria (see Section 2.3) were excluded at this stage.

For the second stage, full-text articles were retrieved for all potentially relevant studies, including those lacking abstracts but with titles suggesting relevance, and those with abstracts that did not clearly specify study design or sample characteristics. Each full-text article was then comprehensively evaluated against the previously defined inclusion and exclusion criteria. This thorough approach ensured that inclusion decisions were based on complete information about study methodology and participant characteristics.

### 2.3. Study Eligibility

Studies were eligible for inclusion if they were (a) published in peer-reviewed journals and written in English to ensure terminological consistency, methodological rigor, and reproducibility. The review focused on (b) empirical studies involving original primary data collection, including both quantitative and qualitative research design. Non-empirical works–such as reviews, meta-analyses, book chapters, clinical guidelines, editorials, and comments–were excluded. Likewise, case reports were excluded due to their limited generalizability, methodological heterogeneity, and potential for selection bias. Eligible studies had to (c) assess symptoms associated with SM, either explicitly or implicitly. Acceptable methods of symptom assessment (d) included clinical or behavioral observations, diagnostic interviews, validated questionnaires, or physiological measures. To ensure diagnostic validity (e), only those studies were included in which the entire sample consisted of children or adolescents (aged 0 to 18 years) with a clinical diagnosis of SM. Diagnosis had to be established through one or more of the following criteria: confirmation by a clinician, use of standardized diagnostic instruments or screening tools with established cutoff values, or explicit reference to DSM or ICD diagnostic criteria. Studies were also included if participants were reported to be undergoing treatment specifically for SM. Studies in which only a portion of the sample met diagnostic criteria for SM were excluded, as symptom data could not be reliably attributed to the clinically diagnosed subsample. Studies were further excluded if SM was only a peripheral or secondary topic, or the sample was composed exclusively of non-clinical or community populations.

### 2.4. Data Extraction

After full-text assessment, studies meeting the inclusion criteria were selected for final analysis. Data extraction was performed using a standardized form developed in Microsoft Excel. For each study, the following information was extracted: (a) study design (e.g., case–control, cross-sectional, qualitative), (b) participant characteristics (e.g., sample size, age, gender), (c) diagnostic procedures used to confirm SM, (d) informant type (e.g., caregiver, teacher, clinician), (e) assessment methodology (e.g., standardized questionnaires, diagnostic interviews, behavioral observation, physiological measures), (f) reported symptom profile (e.g., social withdrawal, anxiety, oppositional behavior), (g) study quality indicators, and (h) symptom domains assessed by the authors (e.g., anxiety-related symptoms, behavioral problems) to provide context for the findings and to address possible assessment bias in symptom detection.

### 2.5. Symptom Definition and Coding

A clearly defined coding scheme was applied to ensure a consistent and valid identification of symptoms in the heterogeneous sample of studies included. The conceptual basis for determining what constitutes a symptom was guided by the definition proposed by [60] ([60]), who describe symptoms as enduring or recurring subjective experiences or behavioral patterns indicative of mental health problems. This definition encompasses both externally observable and self-reported phenomena, including emotional, behavioral, or physiological expressions of dysfunction.

Consistent with this definition, features were coded as symptoms if they reflected a clinically relevant deviation from typical development and were explicitly measured at the time of assessment. Features described as premorbid, such as early childhood temperament traits, were excluded. Likewise, formally diagnosed comorbidities were not coded as symptoms, as the review aimed to isolate symptom characteristics of SM itself.

To account for methodological diversity and varying evidentiary standards across research, all studies included were classified into two main categories based on the application of normative anchors. Category 1 included studies using normative anchoring that employed standardized tests, with existing normed data or standardized cutoff values. Category 2 comprised studies without explicit normative references, but with qualitative descriptions, clinical observations, or non-standardized measures lacking established normative benchmarks. Studies were only classified as norm-referenced (Category 1) if normative data, cut-off scores, or standardized comparisons were explicitly reported in the results. Studies using standardized instruments solely for diagnostic purposes without reporting normative comparisons were categorized as non-norm-referenced (Category 2) to ensure consistency and avoid inferring unreported information.

This differentiation allowed for a more nuanced interpretation of the clinical significance of reported symptoms. Coding criteria were adapted to suit different study designs.

#### 2.5.1. Quantitative Studies Using Standardized Instruments

Within quantitative studies using standardized instruments, such as diagnostic interviews or psychometric questionnaires, symptoms were coded when the reported scores exceeded one standard deviation from normative means. This method aimed to sensitively identify potential symptoms related to SM. When no normative data were available, coding relied on cutoff values defined by the original study authors. This ensured consistency in identifying clinically significant deviations while accommodating study-specific reporting formats.

#### 2.5.2. Case–Control Studies with Healthy Controls

In case–control studies comparing children with SM to healthy controls, symptoms were only coded if the SM group exhibited significantly higher levels of the feature in question or exceeded the one standard deviation threshold on standardized measures, assuming such differences indicate meaningful clinical features of SM.

#### 2.5.3. Studies Comparing Clinical Populations

In contrast, case–control studies comparing SM groups to other clinical populations, such as children with anxiety or emotional disorders, were treated more conservatively. Given that both groups are likely to exhibit elevated symptom levels, these comparisons provide limited insight into symptom specificity. Accordingly, symptoms were only coded in these studies when scores for the SM group exceeded the clinical thresholds defined by instruments or standard deviations from normative means. Descriptive or non-standardized data alone were not considered sufficient in this context.

#### 2.5.4. Qualitative Studies and Clinical Case Notes

In qualitative studies and those based on medical case notes, symptoms were coded only if clearly described by informants, clinicians, or researchers and if they indicated observable difficulties relevant to SM.

#### 2.5.5. Intervention Studies

In the case of intervention studies, only baseline (pre-treatment) values were considered, as the goal was to capture symptom characteristics prior to therapeutic input. Post-treatment changes were not included in the coding process. Baseline symptoms were only coded if they met the same criteria of clinical relevance as outlined above.

#### 2.5.6. Interrater Reliability of Symptom Rating

To ensure reliability of symptom extraction and coding, both authors independently extracted and coded symptoms from randomly selected subsets of studies. For studies with normative anchoring (Category 1), 10 studies were randomly selected for dual coding, while another 10 studies were randomly selected from studies without direct normative reference (Category 2). Interrater reliability was substantial for both study categories (Category 1: *κ* = 0.89; Category 2: *κ* = 0.86). Any discrepancies in symptom coding were resolved through discussion until consensus was reached.

### 2.6. Methodological Quality Assessment

Methodological quality of included studies was evaluated using the critical appraisal tools provided by the Joanna Briggs Institute (JBI; [3]). The JBI toolkit provides specific checklists adapted to different study designs and methods, with items rated as “yes”, “no”, “unclear”, or “not applicable”. For each study, the most appropriate checklist was selected based on its study design. Quality ratings were calculated as the percentage of applicable items rated “yes” to allow for comparability across tools with differing numbers of items. Studies were classified as low quality (<50% “yes” ratings), moderate quality (50–79%), or high quality (≥80%).

One author performed quality appraisal for all included studies. To assess interrater reliability, a second reviewer independently rated a random subset comprising 10 studies. Agreement between raters was substantial (*κ* = 0.60).

### 2.7. Data Analysis

Data analysis followed a descriptive, integrative approach aimed at systematically mapping the symptomatology of SM across diverse study types and assessment methods. Due to the variety of methodological approaches in the studies included and the exploratory character of this review, a descriptive strategy was considered most appropriate to provide a comprehensive overview of the existing evidence. All statistical analyses were performed using R Statistical Software (v3.3; [40]).

To organize the large number of individual features reported in the original studies, symptoms were grouped into broader clusters in a stepwise process. First, all symptom labels as reported by study authors were extracted verbatim. Features with similar content but varying terminology (e.g., “aggression”, “externalizing behavior”, or “oppositional behavior”) were compared and evaluated for conceptual overlap. Then, these features were integrated into broader symptom clusters when they were judged to reflect the same underlying clinical phenomenon. Clustering was conducted by the first author and guided by established diagnostic frameworks, patterns in the data, and relevant literature. A detailed overview of the final symptom clusters and their encompassed features is provided in Appendix A.

To enhance interpretation of symptom findings, we examined whether a given symptom was (a) assessed and (b) identified as clinically relevant in each study. This allowed us to distinguish between symptoms that were not evaluated at all and those that were analyzed but remained within normative ranges, providing essential context for interpreting both positive and negative findings across studies.

In cases where a study incorporated multiple informants or assessment techniques (e.g., parent and self-reports) that recognized the same symptom, that symptom was included just once in the main descriptive summaries for each study to prevent an artificial inflation in symptom prevalence. At the same time, in order to account for the potential differences among informants and measurement methods, counts specific to each method were preserved and displayed separately.

The analysis was divided into two parts based on whether studies applied any form of normative reference to account for differences in evidentiary standards (see Section 2.5). This stratification enabled a more refined interpretation of the clinical significance of reported symptoms.

## 3. Results

### 3.1. Study Selection

Figure 1 presents a PRISMA flowchart illustrating the study selection process. A total of 3946 records were identified across all databases. After removing 463 duplicates, 3483 records remained for title and abstract screening based on the predefined inclusion criteria. Full texts were sought for 227 records, including those without an abstract but with titles suggesting potential relevance to the research question, as well as records with abstracts that did not clearly specify study design or sample characteristics. Thirteen full texts could not be retrieved, despite efforts to contact the corresponding authors. Consequently, 214 reports were assessed for eligibility. A total of 82 studies met inclusion criteria and were included in the final synthesis.

### 3.2. Characteristics of Included Studies

Appendix A presents key characteristics of the studies included in this review. The final sample comprised 82 studies published between 1979 and 2025. Most participants were female (63.16%; *n* = 9 studies did not report on gender), with an average age of 8.43 years (*SD* = 2.45; *n* = 18 studies did not report mean age values). In most cases, the diagnosis of SM was based on DSM-based criteria. Study designs included cross-sectional, case–control, cohort, qualitative, and mixed-method approaches.

### 3.3. Risk of Bias and Quality Assessment

Methodological quality of included studies was evaluated using the relevant JBI tools, according to study designs. Overall, 59.8% of the studies were rated moderate quality, 34.1% high quality, and 6.1% low quality. Common limitations included missing information on identifying and managing confounding factors. Appendix A presents the quality ratings for each study, providing an overview of the methodological strengths and limitations identified across the included studies.

### 3.4. Synthesis of Results: Studies with Normative Reference

Out of all 82 included studies, 63 incorporated some form of normative reference–defined as the use of standardized instruments with validated norms or cutoff-values or comparisons to typically developing control groups. Across these studies, a total of 37 symptom clusters were identified.

The most frequently reported symptoms included social anxiety (clinically elevated in *n* = 31 studies; 49.2%), non-specific anxiety (*n* = 29; 46%), internalizing problems (*n* = 14; 22.2%), withdrawal (*n* = 12; 19.0%), depressive symptoms (*n* = 11; 17.5%), as well as impaired social interaction skills and impaired social/peer relationships (both *n* = 8; 12.7%).

The majority of these symptoms were assessed using parent or caregiver questionnaires (*n* = 47; 74.6%), followed by self-report questionnaires (*n* = 21; 33.3%), teacher-report questionnaires (*n* = 15; 23.8%), physiological measures (*n* = 9; 14.3%), clinical observations (*n* = 6; 9.5%), standardized motor, language, or cognitive tests (*n* = 5; 7.9%), and clinician-rated assessments (*n* = 2; 3.2%).

An overview of all identified symptoms and their method-specific frequencies is provided in Figure 2. Since a single study may have multiple informants or methods identifying the same symptom, the total number of observations in the figure may exceed the number of studies that reported that symptom.

Table 1 summarizes how many studies with normative reference (Category 1) assessed each symptom (i.e., used instruments targeting the symptom) and how often the symptom was reported as clinically relevant. Percentages reflect the proportion of studies with a positive finding among those that assessed the symptom.

### 3.5. Synthesis of Results: Studies Without Normative Reference

In the remaining 19 studies without direct normative reference, symptom identification relied primarily on clinical interviews, narrative case descriptions, or behavioral observations. These studies often described symptoms not captured in standardized tools, thereby contributing complementary insights into the symptomatology of SM.

The most frequently described symptoms in this group included externalizing and behavioral problems (*n* = 10; 52.6%), anxiety (*n* = 9; 47.4%), shyness (*n* = 8; 42.1%), communication problems (*n* = 7; 36.8%), and depressive symptoms (*n* = 6; 31.6%). Additionally, less frequently reported symptoms are listed and illustrated in Figure 3.

Symptom assessment in these studies was mainly conducted using clinical observations (*n* = 7; 36.8%), parent or caregiver reports (*n* = 4; 21.1%), self-reports (*n* = 2; 10.5%), teacher reports (*n* = 2; 10.5%), and standardized language or developmental assessments (*n* = 2; 10.5%).

Figure 3 summarizes the full range of symptom clusters described in these studies, differentiated by type of assessment and informant.

Unlike studies with normative reference, these reports rarely relied on standardized symptom assessment tools. Consequently, no distinction was made between assessed and clinically significant symptoms. Instead, all symptoms included in this category reflect features explicitly described as present by the respective informants or authors.

## 4. Discussion

### 4.1. Main Findings: SM as a Multisymptomatic Disorder

This review is the first to systematically examine the symptomatology of SM across diverse empirical studies. Since there have been no previous syntheses in this domain, our approach was necessarily exploratory and narrative in nature and aimed to map the landscape of symptoms reported in the SM literature. Instead of estimating prevalence rates, this review sought to identify which symptoms have been studied and reported in association with SM and under what methodological conditions they appeared.

Our findings challenge the current diagnostic conceptualization in the DSM-5 criteria of SM as defined solely by failure to speak in specific situations. Instead, the evidence observed reveals a complex, multisymptomatic condition involving emotional, behavioral, and social features that extend well beyond mutism. These results are consistent with earlier observations and with research that has questioned the reduction in SM to failure to speak alone ([23]; [44]; [59]).

Social and general anxiety were among the most frequently assessed and identified symptoms across studies, especially in those utilizing standardized assessments. Notably, nearly every study that evaluated anxiety-related symptoms found them to be clinically relevant–for instance, all studies assessing social anxiety reported its presence. While this consistency reflects the current conceptualization of SM as an anxiety disorder, the underlying mechanisms remain to be clarified. Instead, it is important to recognize that symptom frequency does not necessarily indicate clinical significance or etiological causality. The consistent documentation across studies tells us what has been observed, but not why these symptoms occur or how they relate to each other. For instance, the presence of social anxiety in SM could reflect multiple underlying scenarios. Understanding these relationships requires longitudinal and experimental approaches that extend beyond the cross-sectional symptom documentation available in most current studies.

Current evidence nevertheless indicates that anxiety is by far the most prominent feature in the majority of children with SM, a finding consistently reported in research and reflected in the present review. However, it is not the only feature. The frequent co-occurrence of additional symptoms—including withdrawal, depressive symptoms, social skills deficits, and behavioral difficulties—suggests that anxiety alone does not fully account for SM’s clinical presentation. The documentation of additional externalizing behaviors presents an intriguing yet complex finding that requires cautious interpretation. While clinical reports and qualitative accounts often describe stubborn and oppositional behaviors in children with SM, these features are rarely confirmed in standardized assessments, despite many studies using normative instruments to capture them. [46] ([46]) argue that defiance-like behaviors in SM might represent situational protest reactions rather than stable traits. They hypothesize that these behaviors might be triggered by acute anxiety and mistaken by interactional partners as deliberate opposition. The discrepancy between clinical observations and standardized assessments suggests that current assessments may not adequately capture the contextual and reactive aspects of these kinds of behaviors in SM. Nonetheless, the consistent documentation of oppositional behaviors across clinical contexts indicates they represent meaningful clinical phenomena that warrant systematic investigation. The hypothesis of a distinct subgroup within the SM population characterized by elevated externalizing symptoms remains worth exploring, as this could have important implications for both diagnostic classification and treatment approaches ([1]; [16]).

### 4.2. The Assessment Gap: What We Miss and Why

The recognition that symptom profiles differ between qualitative observation and standardized measurement points to a broader methodological issue in SM research–one that this review systematically addresses. By comparing these approaches, we demonstrate that apparent contradictions in previous research actually reflect complementary perspectives on SM’s clinical complexity. From an empirical standpoint, our findings clearly indicate that anxiety represents the central phenomenon in most children with SM, being consistently documented across studies and assessment approaches. However, the variation in symptom reporting depending on assessment approach and informant type reveals that other clinically relevant aspects beyond anxiety have been systematically underrepresented in the literature. This discrepancy does not challenge the primacy of anxiety in SM but suggests that the symptom landscape may be broader than what is typically represented in the literature.

Studies that relied on standardized tools primarily captured internalizing symptoms, especially those related to anxiety. In contrast, qualitative studies and clinical observations without normative reference revealed a broader array of emotional and behavioral difficulties. The structure and intent of most standardized instruments—designed primarily to assess anxiety—have likely contributed to this potentially narrowed focus. As a result, clinically meaningful features beyond anxiety may have been systematically underreported, not because they are absent, but because they fall outside the scope of the measurement tools commonly used. Thus, the strong emphasis on anxiety in SM research, while valid and foundational, may also reflect prevailing assessment practices that limit our ability to capture the full complexity of the disorder.

While our findings highlight that standardized assessment tools may underrepresent symptom dimensions beyond anxiety, it is important to acknowledge that standardized instruments offer high reliability, comparability, and provide quantifiable data that are essential for establishing symptom prevalence and guiding evidence-based treatments. To capture the full symptom spectrum of SM, standardized measures should be complemented with qualitative assessments and open-ended clinical observations. Future research should therefore aim at refining existing tools, broadening item content to include social, emotional, and behavioral aspects.

The type of informant further influenced symptom identification patterns. Consistent with previous findings from original studies, parents emphasized internalizing problems and withdrawn behavior ([9]; [24]; [29]), while externalizing and oppositional traits were more frequently reported by clinicians (sometimes relying on parental reports) in medical case reports. Notably, child self-reports were scarce among the included studies. This underrepresentation of children’s own experiences is particularly striking, especially since earlier research has demonstrated that children with SM often report lower levels of anxiety than clinicians or parents observe ([63]; [64]) These findings underscore that a comprehensive assessment of SM requires the perspectives of multiple informants and various methodological approaches to effectively capture its inherently context-dependent nature.

More broadly, our findings suggest that research and informant expectations may shape what is documented in SM studies. The field’s focus on anxiety-based instruments may create a cycle where expected symptoms are consistently found while other domains remain underexplored, and different informants may selectively report symptoms that align with their understanding of the disorder.

### 4.3. Implications for Diagnosis and Classification

Our findings raise important questions about the adequacy of current diagnostic frameworks for SM. While the DSM-5 conceptualizes mental disorders as syndromes involving multiple domains of dysfunction, SM remains defined by a single observable behavior in the diagnostic criteria. Our findings suggest that failure to speak, while central, may represent only the most visible symptom in a broader constellation of emotional and behavioral difficulties.

The heterogeneity of symptoms observed across studies aligns with previously proposed hypotheses about SM subtypes ([10]; [17]; [33]). Externalizing and oppositional behaviors were noted in several studies, particularly those without normative anchoring, raising the possibility of a subgroup of children with SM characterized by affective dysregulation or behavioral difficulties ([1]; [16]). We regard this pattern as preliminary observations, underscoring the need for systematic research to clarify whether such features represent a meaningful subgroup of SM or reflect comorbidity and contextual influences.

A more differentiated, symptom-based diagnostic approach in SM evaluation could reduce the risk of neglecting clinically relevant features beyond failure to speak. This seems especially relevant given the high rates of undetected cases in children with SM ([49]) and the long average delay between symptom onset and clinical presentation ([41]). To advance a more nuanced understanding of SM and refine existing diagnostic criteria, there is a clear need for systematic research that captures the full spectrum of symptoms associated with the condition across diverse assessment methods and informant perspectives.

### 4.4. Implications for Clinical Practice

From a clinical perspective, our findings underscore the need for broader diagnostic assessments in SM. Evaluations should extend beyond simply confirming failure to speak and include structured as well as open-ended assessments that examine internalizing symptoms, social functioning, and emotion regulation. Furthermore, clinicians should consider gathering information from diverse sources. The variation in symptom reporting across parent, teacher, clinician and self-report measures demonstrates that each perspective contributes unique insights into the child’s functioning across different contexts, which is particularly relevant for a disorder whose symptomatology is highly influenced by situational factors. Reconciling such discrepancies represents a central clinical challenge. Rather than relying on a single perspective, best practice involves systematically comparing reports and integrating them with clinical observations.

The diversity of symptom expressions in SM also cautions against uniform treatment protocols. Current therapeutic approaches have primarily focused on reducing anxiety, and results from randomized controlled trials demonstrate that cognitive-behavioral therapy effectively benefits children with SM, showing improvements in social anxiety, speaking behavior, and overall functioning ([4]; [12]; [37]). Nevertheless, research remains limited on children exhibiting wider symptom profiles, including oppositional conduct or emotional dysregulation. For these children, additional interventions–like parental training, emotion regulation strategies, or medication–should be considered to better address the needs of children who do not respond to anxiety-focused treatments alone ([63]).

### 4.5. Future Research Directions

This review identified several gaps and challenges in current SM research that need to be addressed in future studies. Given the limitations in SM assessment discussed above, future studies should employ comprehensive, multidomain assessment tools to better capture the different aspects of SM beyond its anxious core. Consistent with previous research ([18]; [63]; [64]), we recommend using multi-informant approaches to account for variability in symptom reports from parents, teachers, and clinicians. Developing and implementing age-appropriate, indirect, or nonverbal self-report measures, such as pictorial scales ([32]), may be particularly important. Recent reviews have emphasized this need ([43]), as child-reported data provide crucial firsthand perspectives that complement external observations.

Two areas emerge as especially promising for improving our theoretical understanding of SM. First, research should systematically explore the mechanisms underlying SM’s diverse symptom presentations, as it remains largely unclear why children exhibit certain symptom patterns ([57], [58]). This mechanistic focus should involve exploring how SM can be distinguished from closely related conditions–such as social anxiety disorder or autism spectrum disorder–to improve diagnostic precision ([8]; [32]; [39]; [47]; [55]). Second, despite their methodological limitations, case studies warrant systematic inclusion in future syntheses. Our findings suggest that case-based evidence frequently identifies behavioral and affective symptoms not captured by standardized tools, potentially offering valuable insight into rarely assessed symptom dimensions.

Finally, the meaningful heterogeneity observed within SM builds on promising initial work exploring SM subtypes ([10]; [17]; [33]). These foundational studies provide valuable groundwork that should be expanded through more comprehensive symptom assessment and advanced statistical approaches such as latent class analysis. Future subtyping efforts should also consider potential age and gender-related differences in symptom presentation. Such expanded investigations could derive more nuanced symptom-based subgroups, with important implications for both diagnostic classification and treatment planning.

Such symptom-based approaches align with contemporary psychopathology frameworks that emphasize dimensional models over categorical boundaries ([27]), transdiagnostic perspectives examining shared mechanisms across disorders ([15]), and network approaches conceptualizing mental disorders as interconnected symptom systems ([7]; [42]). Applied to SM, these frameworks may facilitate a more nuanced understanding of symptom patterns and their interrelationship.

### 4.6. Limitations and Interpretive Considerations

Several methodological factors limit the generalizability of our findings and require careful consideration. We deliberately included studies that used different research methods to gather all available evidence on SM symptoms–a field that has not yet been systematically reviewed. Although differences in study design, measurement tools, and data sources complicate interpretation and may have introduced potential biases, they also offer a more comprehensive picture than restricting analysis to only highly standardized studies. The quality assessment conducted revealed that most studies were of moderate to high quality, with common limitations mainly related to the control of confounding variables.

Most importantly, the aim of this review was not to provide epidemiological prevalence rates but to map the current research landscape. Reported frequencies reflect how certain symptoms have been studied and found–not how common they are among children with SM. We used study-level data and did not apply sample size weights, meaning that small and large studies contributed equally. This approach promotes inclusivity but limits statements about symptom typicality and severity.

Another central factor to consider is the influence of comorbid disorders. Many studies included participants with additional psychiatric diagnoses, making it difficult to distinguish symptoms specifically attributable to SM from those caused by co-occurring disorders. This is particularly relevant given the high comorbidity between SM and social anxiety disorder. The conceptual overlap between these disorders makes it difficult to identify symptoms unique to SM and again underscores the need for a more defined diagnostic differentiation ([19]; [39]; [58]). In this review, comorbidities were not coded as symptoms to maintain a symptom-based perspective; however, their presence may have affected the reporting of symptoms within the studies included.

Furthermore, it is important to note that our review did not focus on developmental precursors, vulnerabilities, or etiological factors. While previous studies have examined links between SM and aspects such as language delay ([28]; [51]) or motor development ([30]), our focus was on current symptom presentations at the time of assessment. As a result, we did code symptoms such as attention or cognitive problems but did not include broader developmental constructs like general intelligence.

Studies also differed substantially in how they reported symptoms. Some of them used broadband diagnostic scales (e.g., “internalizing problems”) without further decomposition into more fine-grained symptoms, thereby creating conceptual overlap among certain groups of symptom clusters. Additionally, a wide variety of standardized instruments was used to assess symptoms–often differing in scope, item content, and theoretical underpinnings. As a result, it remains unclear whether studies that nominally assessed the same construct (e.g., anxiety) were capturing equivalent psychological phenomena, reducing comparability across studies.

Our coding approach included symptom reports that were one standard deviation above mean values. While this allowed us to capture a wide range of symptom expressions, it may have included milder symptoms that are not necessarily clinically significant. However, this broad approach was chosen to ensure we did not miss potentially important symptoms.

Finally, both the extraction and the grouping of symptoms inevitably involved some degree of subjective interpretation, particularly for qualitative data. In our review, the clustering of individual features into broader categories was conducted by one author, guided by diagnostic frameworks and coherence across the data. Although subsets of extracted symptoms were cross-checked, some risk of interpretive bias cannot be entirely ruled out. Nonetheless, our high interrater reliability demonstrates that consistent and reliable coding was achievable despite these challenges, strengthening confidence in our conclusions.

## 5. Conclusions

To the best of our knowledge, this review provides the first systematic synthesis of symptomatology in SM. The results show that SM extends beyond failure to speak alone and often includes anxiety, affective symptoms, social difficulties, and, sometimes, behavioral dysregulation. This complexity is not adequately reflected in current diagnostic frameworks and may be underestimated in commonly used standardized assessments. Our results emphasize the need for broader, multi-informant assessment approaches and more inclusive diagnostic criteria that capture the full spectrum of symptoms related to SM. By documenting the range of symptom patterns reported across different empirical studies, this systematic review highlights key areas where future research is needed. Ultimately, a more differentiated understanding of SM can improve identification and guide tailored interventions, ensuring that children with SM receive the recognition and support they need.

## Figures and Tables

**Figure 1 behavsci-15-01485-f001:**
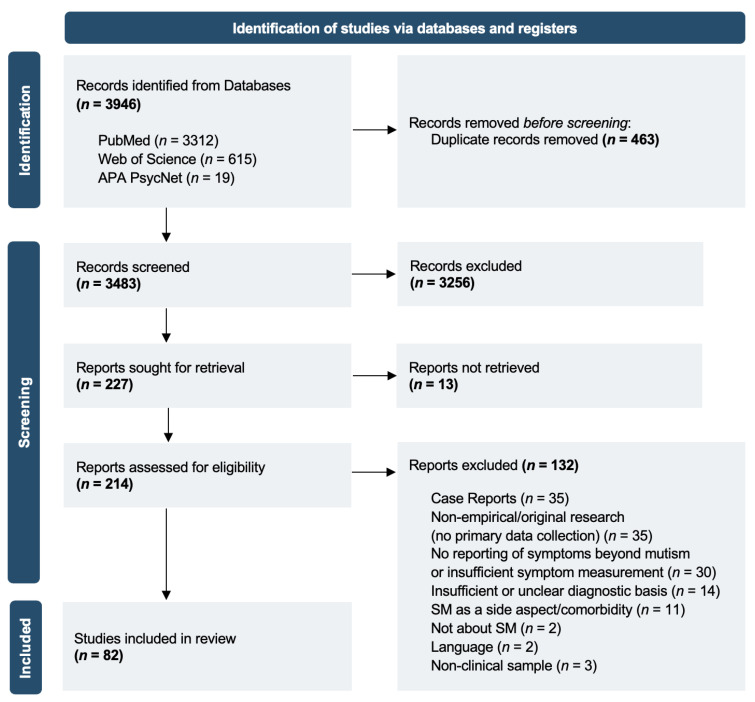
PRISMA flowchart of the study selection procedure.

**Figure 2 behavsci-15-01485-f002:**
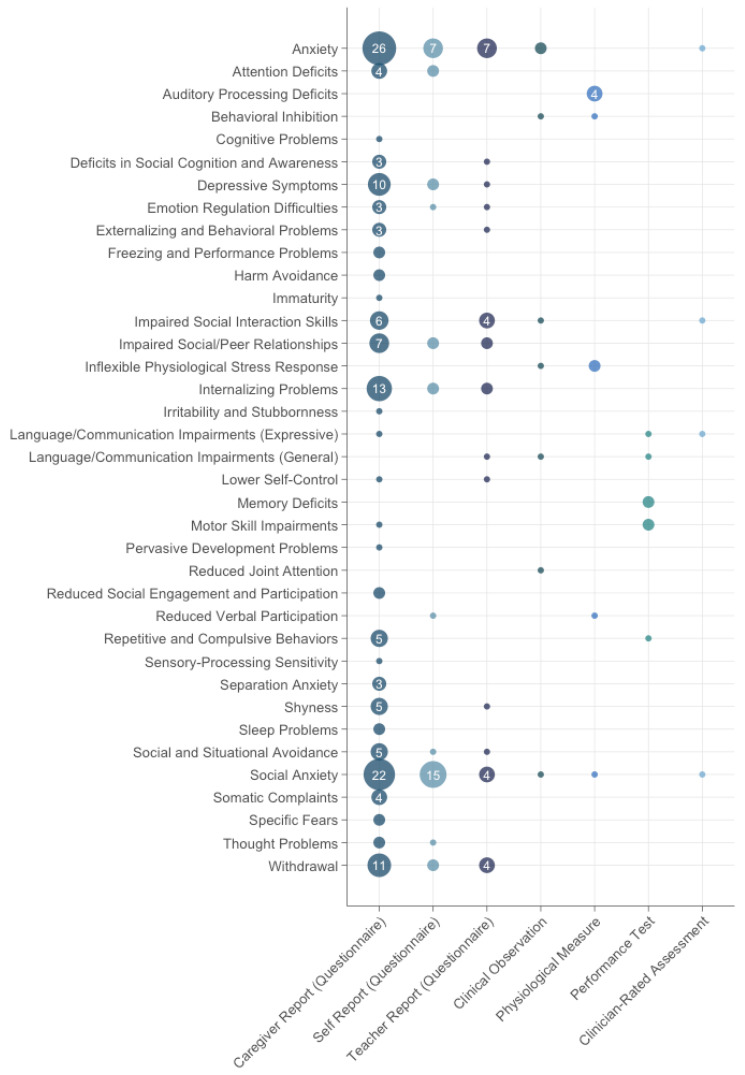
Frequency of symptom reports by assessment method in studies with normative reference (Category 1). Circle size indicates the number of method-specific symptom observations.

**Figure 3 behavsci-15-01485-f003:**
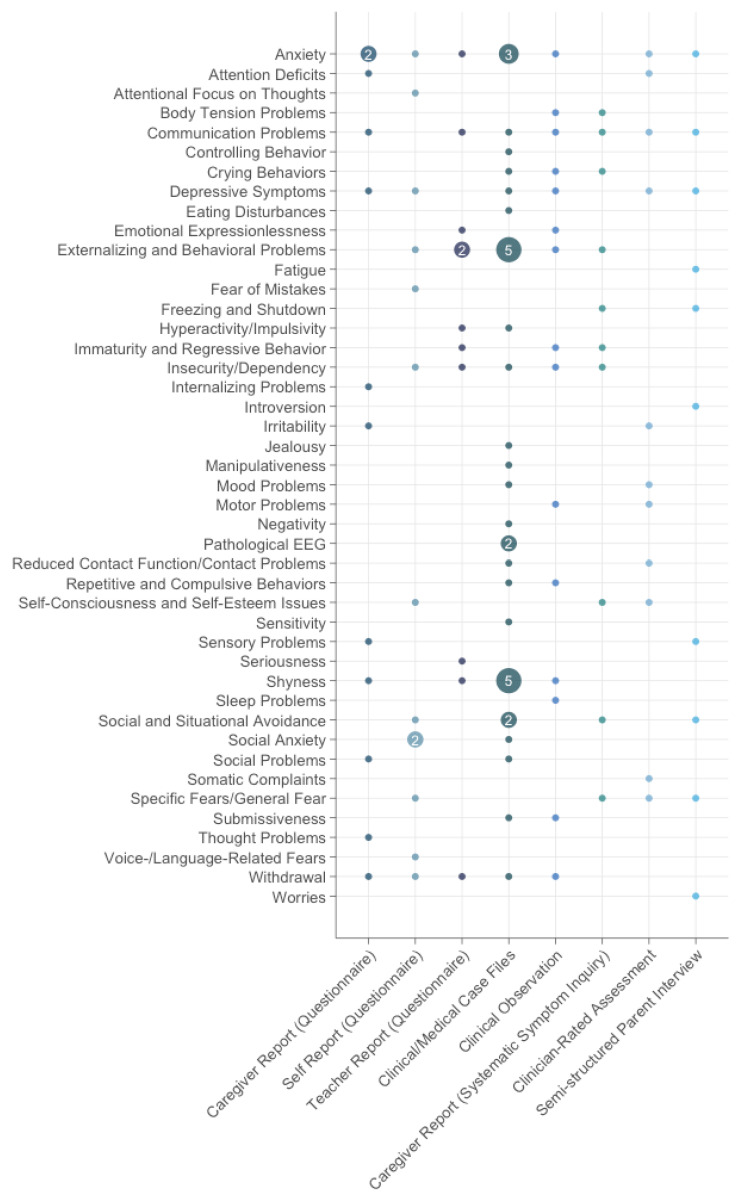
Frequency of symptom reports by assessment method in studies without normative reference. Circle size indicates the number of method-specific symptom observations.

**Table 1 behavsci-15-01485-t001:** Symptom Assessment and Reported Findings in Studies with Normative Reference.

Symptom Cluster	Assessed	Identified as Clinically Relevant	% Identified as Clinically Relevant
Anxiety	35	29	82.9
Attention Deficits	17	5	29.4
Auditory Processing Deficits	4	4	100.0
Behavioral Inhibition	2	2	100.0
Cognitive Problems	4	1	25.0
Deficits in Social Cognition and Awareness	3	3	100.0
Delinquency	6	0	0.0
Depressive Symptoms	14	11	78.6
Dissociation	1	0	0.0
Emotion Regulation Difficulties	7	5	71.4
Externalizing and Behavior Problems	24	3	12.5
Freezing and Performance Problems	2	2	100.0
Harm Avoidance	3	2	66.7
Immaturity	2	1	50.0
Impaired Social Interaction Skills	8	8	100.0
Impaired Social/Peer Relationships	11	8	72.7
Inflexible Physiological Stress Response	3	3	100.0
Internalizing Problems	14	14	100.0
Irritability and Stubbornness	2	1	50.0
Language/Communication Impairments (Expressive)	5	3	60.0
Language/Communication Impairments (General)	5	2	40.0
Language/Communication Impairments (Receptive)	7	0	0.0
Lower Physical Competence	1	0	0.0
Lower Self-Control	2	1	50.0
Impaired Mathematic Competence	2	0	0.0
Memory Deficits	5	2	40.0
Motor Skill Impairments	2	2	100.0
Negativism	1	0	0.0
Pervasive Developmental Problems	1	0	0.0
Reduced Prosocial Behavior	2	0	0.0
Altered Pulse Rate	1	0	0.0
Impaired Reading Competence	1	0	0.0
Reduced Academic Performance	1	0	0.0
Reduced Joint Attention	1	1	100.0
Reduced Social Engagement and Participation	1	1	100.0
Reduced Verbal Participation	5	2	40.0
Repetitive and Compulsive Behaviors	7	6	85.7
Restlessness/Impulsivity	1	0	0.0
Sensory-Processing Sensitivity	1	1	100.0
Separation Anxiety	4	2	50.0
Sleep Problems	2	2	100.0
Reduced Sociability	1	0	0.0
Social and Situational Avoidance	5	5	100.0
Social Anxiety	31	31	100.0
Somatic Complaints	10	3	30.0
Specific Fears	2	2	100.0
Inflexibility	1	0	0.0
Reduced Persistence	1	0	0.0
Thought Problems	6	2	33.3
Withdrawal	12	12	100.0

## Data Availability

No new primary data were generated in this review. Extracted data tables and coding sheets compiled during the review process are available upon request.

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
