# Peer review of "What Can We Learn from the Previous Research on the Symptoms of Selective Mutism? A Systematic Review"

_behavsci, 2025, doi:10.3390/bs15111485_

Round 1
Reviewer 1 Report
Comments and Suggestions for Authors
Overall, I think the paper „What can wie learn from the previous research on the symptoms of selective mutism? A systematic review“ done by Kleinheinrich and Vogel can make a contribution to the literature. The following suggestions may help to improve the paper on a number of specific aspects:
Introduction:
L 47ff: The effects of selective mutism cannot be viewed solely in functional terms. The uncertainty and psychological stress caused by the symptoms are particularly significant aspects that can contribute to secondary symptoms.
L 77ff: The introduction creates an expectation that cannot be fulfilled on the basis of the present study: The expectation that the diagnostic criteria for selective mutism could be complemented requires a broader research approach than the present one. This aspect should therefore not be addressed in the introduction, but rather in the discussion as a perspective: What is lacking in previous research to be able to clarify the diagnostic criteria?
The introduction is very narrowly focused on SM as an anxiety disorder. The introduction lacks an overview of explanatory approaches to selective mutism that deviate from the currently most commonly used concept of an anxiety disorder underlying selective mutism: For example, associations with ASD, SM as a stress-related disorder, associations with trauma, and SM symptoms as dissociative reactions should be presented, as they are part of previous research.
The study by Driessen et al. (2020) should also be cited, with its conclusion that not all children with selective mutism exhibit anxiety symptoms, and thus the classification of the disorder as an anxiety disorder was premature. This paper also addresses what current research cannot yet answer.
Materials and Methods
The chosen research method is not appropriate for answering the question posed in the title of the review. A thorough understanding of the disorder's symptomatology can only be achieved through a broad search of available articles. Case reports, for example, are of great value, as they are not subject to the constraints of expectations. Even if this limitation is addressed in the discussion, it should be made clear in the abstract and subtitle.
A list of the diagnostic instruments used in the respective studies should be added to assess which symptoms could be reported at all.
The calculation of interrater reliability is based on only 10 studies.
Results
The results should be presented in content-based, not alphabetical order (Figure 2 and Table 1).
The authors should also consider a possible relation between the recorded symptoms and children’s age, and if possible, gender.
Discussion:
Lines 519-520: The frequency of a symptom does not determine its significance. The statement "All studies assessing social anxiety reported its presence. This consistency supports the classification of SM as an anxiety disorder in current diagnostic manuals" is scientifically incorrect. There are very different interpretations of why a symptom occurs frequently. The assumption that social anxiety, for example, occurs secondary to selective mutism, calls into question the significance of anxiety in the development of the disorder: It thus questions the connection between the frequency of a symptom and causality and significance.
The finding that research is often conducted in line with expectations and thus important information is not captured because it is not consistent with what the research group expects should be more clearly stated as a result of this study.
The reported symptoms not captured in standardized tools demonstrate the limitations of the research.
The informant aspect should clarify that expectations guide the reporting of symptoms (Rosenthal effect).
Age and gender effects were not taken into account.
Author Response
Comment 1: Introduction: L 47ff: The effects of selective mutism cannot be viewed solely in functional terms. The uncertainty and psychological stress caused by the symptoms are particularly significant aspects that can contribute to secondary symptoms.
Response 1: Thank you for this important addition, we fully agree with the point raised. We have therefore added this suggestion almost verbatim into the introduction (L 69ff), emphasizing that, beyond functional impairments, the uncertainty and psychological stress caused by the inability to speak in expected situations represent a significant burden in themselves.
Comment 2: L 77ff: The introduction creates an expectation that cannot be fulfilled on the basis of the present study: The expectation that the diagnostic criteria for selective mutism could be complemented requires a broader research approach than the present one. This aspect should therefore not be addressed in the introduction, but rather in the discussion as a perspective: What is lacking in previous research to be able to clarify the diagnostic criteria?
Response 2: We agree that our study does not provide a sufficient empirical basis to propose specific changes to the diagnostic criteria of selective mutism. Our intention was to highlight that a more differentiated conceptualization may ultimately support future refinements, not to suggest that our review alone could establish such revisions.
Comment 3: The introduction is very narrowly focused on SM as an anxiety disorder. The introduction lacks an overview of explanatory approaches to selective mutism that deviate from the currently most commonly used concept of an anxiety disorder underlying selective mutism: For example, associations with ASD, SM as a stress-related disorder, associations with trauma, and SM symptoms as dissociative reactions should be presented, as they are part of previous research. The study by Driessen et al. (2020) should also be cited, with its conclusion that not all children with selective mutism exhibit anxiety symptoms, and thus the classification of the disorder as an anxiety disorder was premature. This paper also addresses what current research cannot yet answer.
Response 3: Thank you for this valuable feedback regarding the conceptualization of selective mutism. We agree that a more comprehensive presentation of different theoretical approaches strengthens the introduction and better reflects the current state of the field. We have revised parts of the introduction to provide a more balanced perspective on SM conceptualization. Specifically, we have incorporated the important meta-analysis by Driessen et al.. The revised text appears in the first paragraph of the introduction (L 48ff).
Comment 4: Materials and Methods: The chosen research method is not appropriate for answering the question posed in the title of the review. A thorough understanding of the disorder's symptomatology can only be achieved through a broad search of available articles. Case reports, for example, are of great value, as they are not subject to the constraints of expectations. Even if this limitation is addressed in the discussion, it should be made clear in the abstract and subtitle.
A list of the diagnostic instruments used in the respective studies should be added to assess which symptoms could be reported at all. The calculation of interrater reliability is based on only 10 studies.
Response 4: We appreciate this thoughtful methodological feedback. We fully agree that case reports can provide valuable clinical insights, particularly because they are not bound to the same constraints as standardized instruments and can capture unique or unexpected phenomena. For the purpose of this systematic review, however, we chose to exclude case reports in order to maintain methodological comparability across studies (L 196ff). Case reports may also be prone to selection bias, which limits their generalizability. Including them alongside empirical studies with defined assessment methods would have reduced the overall comparability of findings and the possibility of synthesizing results in a systematic way. We therefore addressed their potential contribution in the discussion, and we agree that they remain an important aspect for future research. We additionally have slightly revised the abstract to make the scope of our review clearer. We now specify that the final synthesis was based in 82 empirical studies with participant samples, thereby clarifying that case reports were not included.
Regarding the diagnostic instruments, we would like to note that a list of the tools utilized in the included studies is already provided in the Appendix/Supplementary Material, where all relevant assessment tools are documented alongside the respective studies. We have ensured that this information is transparent and available to readers, as we agree that it helps provide context for the findings.
In terms of interrater reliability, we note that it was calculated on a random sample of 10 studies, which is in line with methodological standards for systematic reviews. The resulting high agreement indicated that the coding process was performed consistently and reliably.
Comment 5: Results: The results should be presented in content-based, not alphabetical order (Figure 2 and Table 1).
Response 5: We chose to present the results (symptoms) in alphabetical order to make the tables easy to read and allow readers to find specific symptoms quickly. We feel like this structure also avoids unintentionally implying a hierarchy of importance among symptoms. Broader content-based patterns are grouped and interpreted in the discussion section.
Comment 6: The authors should also consider a possible relation between the recorded symptoms and children’s age, and if possible, gender.
Response 6: We fully agree that examining age- and gender-related differences in symptom presentation would be a valuable addition. However, the primary aim of our synthesis was to comprehensively map the range of symptoms reported across the literature, and examining developmental and gender-specific patterns would require a different methodological approach. Additionally, while some of the included studies provided detailed information on participant characteristics, many others offered insufficient detail regarding symptom patterns across different age groups or genders to enable meaningful subgroup analyses. The heterogeneity in how studies reported participant characteristics and symptoms further precluded such analyses.
We acknowledge this as a limitation of our study and have added it to Section 4.5. (L 684ff) as we also consider that examining age- and gender-specific patterns represents a valuable direction for future systematic reviews.
Comment 7: Discussion: Lines 519-520: The frequency of a symptom does not determine its significance. The statement "All studies assessing social anxiety reported its presence. This consistency supports the classification of SM as an anxiety disorder in current diagnostic manuals" is scientifically incorrect. There are very different interpretations of why a symptom occurs frequently. The assumption that social anxiety, for example, occurs secondary to selective mutism, calls into question the significance of anxiety in the development of the disorder: It thus questions the connection between the frequency of a symptom and causality and significance.
Response 7: Thank you for pointing this out. It is absolutely correct that frequency or occurrence does not determine clinical significance or etiological causality, and we should not confuse consistent reporting with causal importance. We have revised the discussion to clarify this crucial distinction. Specifically, we have rephrased the sentence you referenced to read more cautiously. Additionally, we have added a paragraph in section 4.1. (L 534ff), emphasizing that the presence of specific symptoms could reflect multiple scenarios and that determining causality requires approaches that extend beyond cross-sectional symptom documentation.
Comment 8: The finding that research is often conducted in line with expectations and thus important information is not captured because it is not consistent with what the research group expects should be more clearly stated as a result of this study. The reported symptoms not captured in standardized tools demonstrate the limitations of the research. The informant aspect should clarify that expectations guide the reporting of symptoms (Rosenthal effect).
Response 8: Indeed. We have added a paragraph in section 4.2. (L 606ff) that addresses how research and informant expectations may influence what gets documented. We now discuss how the field’s focus on anxiety-based instruments may create a cycle where expected symptoms are consistently found while other domains remain underexplored, and how different informants may selectively report symptoms that fit their understanding of the disorder. This highlights the methodological limitations in current SM research practices and supports the need for more exploratory approaches.
Reviewer 2 Report
Comments and Suggestions for Authors
Research question and rationale: This systematic review addresses an important research question that is relevant to child psychology. The paper challenges the narrow DSM-5 definition of SM and argues convincingly for a multisymptomatic conceptualisation of SM. The research rationale was clearly stated, the objectives were measurable and aligned to the intended scope of the review.
The abstract is informative but could benefit from a clearer structure. Consider using subheadings (e.g., Background, Methods, Results, Conclusion) for readability.
Introduction:
Method:
- Although simple, the search strategy was appropriate and reproducible, and the inclusion/exclusion criteria were unambiguous.
- It would validate the screening process if the paper reported that two reviewers completed full-text screening (phase 2) following initial title and abstract screening (lines 174-175).
- There was a solid rationale for the symptom definition to guide coding features
- Including both studies with normative references and those without broadens the scope of the review.
- Quality assessment,
- The grouping of symptoms into broader clusters based on conceptual similarities - was that done by both authors? Despite being guided by diagnostic frameworks, there is a risk of subjective interpretation bias.
Results:
- The paper synthesises findings from 82 studies, which offer a rich and nuanced understanding of previous diagnoses of SM.
- Well done on an enormous amount of work.
- Clear and logical synthesis and presentation of findings.
Discussion and conclusion:
- Articulated interpretation of results that confirm and support the author's argument.
- Perhaps balance the discussion about the limitations of current standardised tools with their strengths and how they should be improved or complemented.
- The limitations and interpretations were thoroughly discussed. An important point raised is the impact of comorbid disorders and their overlapping influence on symptom presentation. This is a relevant issue across childhood anxiety and related disorders.
- Implications for future research were presented logically and convincingly, presenting several opportunities for further investigations.
Well done on an interesting paper that fills an important gap in the research.
Author Response
Comment 1: Although simple, the search strategy was appropriate and reproducible, and the inclusion/exclusion criteria were unambiguous.
It would validate the screening process if the paper reported that two reviewers completed full-text screening (phase 2) following initial title and abstract screening (lines 174-175).
Response 1: Consistent with established methodological standards for systematic reviews, both title/abstract screening and full-text screening were conducted by a single reviewer. To ensure quality and accuracy, the process was supervised by the last author. In cases of uncertainty, inclusion and exclusion decisions were discussed until consensus was reached.
Comment 2: The grouping of symptoms into broader clusters based on conceptual similarities - was that done by both authors? Despite being guided by diagnostic frameworks, there is a risk of subjective interpretation bias.
Response 2: Thank you for raising this critical methodological issue. We have clarified the procedure of how symptoms were grouped into clusters in Section 2.7. (L 306ff), providing a more detailed stepwise description of the process. While clustering was done by one author, the process was guided by diagnostic frameworks and coherence across the data. In addition, subsets of extracted symptoms were reviewed by both authors, and interrater reliability for symptom coding was found to be substantial, supporting the consistency of our approach. We would also like to kindly refer to Appendix B. There, we have provided an overview listing the symptom clusters, including the associated individual features. At the same time, we fully agree with the reviewer that the clustering process still entails a risk of subjective interpretation, and we have emphasized this limitation in the revised manuscript (Section 4.6.; L 741ff).
Comment 3: Perhaps balance the discussion about the limitations of current standardised tools with their strengths and how they should be improved or complemented.
Response 3: We agree that our discussion should also consider the strengths of standardized instruments. We have therefore revised Section 4.2. (L 587ff) to more explicitly acknowledge their reliability and comparability across studies. At the same time, we highlight that these instruments may best be seen as a valuable foundation that benefits from being combined with qualitative and open-ended approaches. The revised section now includes a paragraph on how current tools can be improved and integrated with broader methods to better capture the full range of SM symptomatology.
Comment 4: The limitations and interpretations were thoroughly discussed. An important point raised is the impact of comorbid disorders and their overlapping influence on symptom presentation. This is a relevant issue across childhood anxiety and related disorders. Implications for future research were presented logically and convincingly, presenting several opportunities for further investigations. Well done on an interesting paper that fills an important gap in the research.
Response 4: We sincerely thank the reviewer for their thoughtful and encouraging feedback. We appreciate the recognition of the comprehensive scope of our review and are pleased that its contribution to the literature was found valuable.
Reviewer 3 Report
Comments and Suggestions for Authors
Thank you for the opportunity to review this manuscript. The authors were original and creative in the conceptualization of this study to determine what other symptoms should be considered within the criteria of selective mutism beyond lack of verbalization. Despite this, I have concerns about the methodology used to determine these additional symptom areas to consider and the conclusions that are made based on these findings. Authors are encouraged to consider the following questions and recommendations to improve the manuscript.
- For the section of studies that did not include normative reference, some of them used questionnaires as well as semi-structured diagnostic interviews - it would seem that these would have normative data to compare to even if that was not included in the paper directly. How then did you determine these did not have norm references? This should be included in the manuscript and then review the data when these are included in the norm-referenced data.
- My biggest concern is that the authors conclude that the symptoms that are deemed as most important clinically to consider as a subtype (i.e., externalizing behavior and behavior problems) come from non-referenced studies and are counted as merely present, but not indicated in the report as being clinically significant by the authors of those particular studies. The authors only highlight this potential subtype from the data that taken from the non-referenced studies. However, the same construct is measured by numerous studies in the referenced studies and is not found to be a significant factor who deem this as relevant. So greater clarification and explanation of why this is most relevant when it is not discussed that a large number of studies did examine externalizing behaviors and did not endorse it being clinically significant per your table. It is not clear how the conclusions were made with both data sets being known.
- Ultimately, the authors are weighing the clinical report where it is unknown what the sample size is (because you did not use weighed sample sizes) and thus it could be only a small number of children in comparison to the studies that might have a larger number of participants.
- It is recommended to use weighted sample sizes so that the data are more accurately represented in terms of importance and significance in what subtypes of selective mutism to consider.
Minor note: in the manuscript Table 1 still shows edits to the table in red
Author Response
Comment 1: For the section of studies that did not include normative reference, some of them used questionnaires as well as semi-structured diagnostic interviews - it would seem that these would have normative data to compare to even if that was not included in the paper directly. How then did you determine these did not have norm references? This should be included in the manuscript and then review the data when these are included in the norm-referenced data.
Response 1: This is a helpful observation which gives us the opportunity to clarify our coding procedure. In our review, a study was only classified as “norm-referenced” if normative data, cut-off scores, or standardized comparisons were explicitly used and reported in the results. While many studies mentioned the use of instruments such as questionnaires or semi-structured diagnostic interviews, they did not always provide corresponding norm-based outcomes. In some cases, instruments were applied solely for diagnostic purposes, without reporting subscale scores or comparisons to control samples. In such instances, we categorized the data as “non-norm-referenced” to ensure consistency and to avoid inferring information not provided by the original publications. We have now clarified this distinction in the manuscript (Section 2.5.; L 242ff).
Comment 2: My biggest concern is that the authors conclude that the symptoms that are deemed as most important clinically to consider as a subtype (i.e., externalizing behavior and behavior problems) come from non-referenced studies and are counted as merely present, but not indicated in the report as being clinically significant by the authors of those particular studies. The authors only highlight this potential subtype from the data that taken from the non-referenced studies. However, the same construct is measured by numerous studies in the referenced studies and is not found to be a significant factor who deem this as relevant. So greater clarification and explanation of why this is most relevant when it is not discussed that a large number of studies did examine externalizing behaviors and did not endorse it being clinically significant per your table. It is not clear how the conclusions were made with both data sets being known.
Response 2: Thank you, this is an important point. As our findings show, the evidence for externalizing and behavioral problems primarily comes from studies without normative reference, whereas standardized assessments have rarely identified these symptoms as clinically significant. We have therefore revised Section 4.3. (L 618ff) to clarify that these findings should not be interpreted as established subtype criteria but rather as preliminary observations emerging from less standardized research. Our intention was not to conclude that externalizing behavior constitutes a definitive subtype of SM, but highlight it as a hypothesis-generating observation that warrants further investigation in future research.
Comment 3: Ultimately, the authors are weighing the clinical report where it is unknown what the sample size is (because you did not use weighed sample sizes) and thus it could be only a small number of children in comparison to the studies that might have a larger number of participants. It is recommended to use weighted sample sizes so that the data are more accurately represented in terms of importance and significance in what subtypes of selective mutism to consider.
Response 3: We appreciate these thoughtful considerations and agree that weighted sample sizes might add even more context to our findings. As noted in our limitation section (4.6.), we acknowledge that our approach means small and large studies contributed equally. However, we believe this was appropriate given our specific aims. Our primary objective was to systematically map the symptom landscape of SM rather than establish prevalence estimates. In this context, the binary question of whether a particular symptom has been identified is methodologically independent of sample size. Our approach provides a comprehensive foundation for understanding symptom breadth, which can inform future research designed to establish weighted prevalence estimates.
Reviewer 4 Report
Comments and Suggestions for Authors
Comments for authors:
This study significantly advances research on Selective Mutism (SM) through rigorous methodology and robust evidence. It demonstrates that the clinical presentation of SM is more complex than captured by current diagnostic standards, challenging the DSM/ICD conceptualization of SM as a "situation-specific absence of speech" and strongly supporting its reconceptualization as a multisymptomatic disorder.
However, several limitations and opportunities for possible improvement are noted as follows:
- The literature search strategy was narrow, relying solely on "selective mutism" and "elective mutism" as keywords, while neglecting terms related to symptom manifestations, comorbidities, and assessment.
- The discussion could be strengthened by exploring shared neurocognitive mechanisms—such as executive dysfunction and social information processing deficits—across SM, social anxiety, and autism spectrum disorders.
- Clinical implications should be more explicitly addressed, particularly strategies for reconciling multi-informant discrepancies (e.g., parent-child ratings) and proposing specific revisions to DSM/ICD criteria, such as incorporating new core or associated symptoms to improve clinical utility.
- The core methodological innovation—comparing standardized tools versus clinical observation—is insufficiently highlighted, especially regarding how it resolves longstanding inconsistencies between these approaches. Furthermore, the role of the findings in facilitating a paradigm shift from a "single-symptom" to a "symptom-network" approach warrants greater emphasis.
Author Response
Comment 1: The literature search strategy was narrow, relying solely on "selective mutism" and "elective mutism" as keywords, while neglecting terms related to symptom manifestations, comorbidities, and assessment.
Response 1: This is a valuable point and gives us the chance to clarify our literature search strategy. We actually decided to use „selective mutism“ and „elective mutism“ as search terms with the intention of being as inclusive as possible. Our rationale was that by using broad diagnostic terms without additional restrictions, we would capture all relevant literature on the topic, regardless of whether studies focused on symptoms, comorbidities, or assessment. We felt that adding more specific terms might actually narrow our search and risk missing studies that used different terminology to describe the same concepts. Since our inclusion criteria required studies to focus specifically on children with a (confirmed) diagnosis of selective mutism, it was appropriate to search explicitly for this condition rather than extending the search in ways that might include studies primarily focused on comorbid disorders where selective mutism was only secondary. This approach allowed us to maintain both comprehensiveness within the literature on selective mutism and precision in our research focus. Importantly, this strategy is consistent with previous systematic reviews and meta-analyses on SM (e.g., Driessen et al., 2020; Koskela et al., 2023).
Comment 2: The discussion could be strengthened by exploring shared neurocognitive mechanisms—such as executive dysfunction and social information processing deficits—across SM, social anxiety, and autism spectrum disorders.
Response 2: We appreciate this thoughtful suggestion. The question of shared neurocognitive mechanisms – particularly executive dysfunction and deficits in social information processing – is undoubtedly significant and warrants increased research attention going forward. However, our focus in this review was to systematically capture and describe the clinical manifestations of selective mutism and a detailed discussion of underlying neurocognitive mechanisms, while highly relevant, would have exceeded the scope and aim of this review.
Comment 3: Clinical implications should be more explicitly addressed, particularly strategies for reconciling multi-informant discrepancies (e.g., parent-child ratings) and proposing specific revisions to DSM/ICD criteria, such as incorporating new core or associated symptoms to improve clinical utility.
Response 3: Thank you for this important suggestion. In response, we have clarified Section 4.4. (L 644ff) to address strategies for dealing with discrepancies between different informants. We now highlight that, rather than prioritizing one informant, best practice involves systematically comparing different reports, integrating them with clinical observations. At the same time, we deliberately refrain from proposing specific revisions to DSM/ICD criteria, as the available evidence is not yet sufficient to justify concrete nosological changes. Instead, we already emphasize in the discussion that future research should evaluate whether incorporating additional/associated symptoms could enhance the clinical utility of diagnostic frameworks.
Comment 4: The core methodological innovation—comparing standardized tools versus clinical observation—is insufficiently highlighted, especially regarding how it resolves longstanding inconsistencies between these approaches.
Response 4: Thank you for this valuable feedback – we agree that this core contribution deserved more prominent highlighting. In response, we have expanded Section 4.2. to better emphasize how our comparison of norm-referenced vs. non-norm-referenced studies represents a novel approach in the SM literature. Specifically, we have enhanced the opening of this Section (L 566ff), stating that our review systematically addresses the discrepancies between assessment approaches, and explaining how this allows us to reconcile apparent contradictions in previous research by showing that these differences reflect complementary rather than conflicting perspectives on SM symptomatology.
Comment 5: Furthermore, the role of the findings in facilitating a paradigm shift from a "single-symptom" to a "symptom-network" approach warrants greater emphasis.
Response 5: We agree that our methodological approach is consistent with broader, contemporary shifts in psychopathology research. In response, we have revised Section 4.5. to emphasize how symptom-based approaches in SM research align with dimensional, transdiagnostic, and network frameworks (L 688ff). While our review was not designed to test such models directly, we highlight that the heterogeneity of symptoms identified in our study may provide a valuable foundation for future studies adopting a symptom-network perspective.
Round 2
Reviewer 3 Report
Comments and Suggestions for Authors
Thank you for your comments and changes in the manuscript to address the concerns. This will be a valuable contribution to the literature.
Reviewer 4 Report
Comments and Suggestions for Authors
Accept